# Accumulation of Advanced Glycation End-Products in the Body and Dietary Habits

**DOI:** 10.3390/nu14193982

**Published:** 2022-09-25

**Authors:** Agnieszka Zawada, Alicja Machowiak, Anna Maria Rychter, Alicja Ewa Ratajczak, Aleksandra Szymczak-Tomczak, Agnieszka Dobrowolska, Iwona Krela-Kaźmierczak

**Affiliations:** 1Department of Gastroenterology, Dietetics and Internal Diseases, Poznan University of Medical Sciences, 61-701 Poznan, Poland; 2Doctoral School, Poznan University of Medical Sciences, 61-701 Poznan, Poland

**Keywords:** advanced glycation end-products, Maillard reaction, diabetes, atherosclerosis, polyphenols, antioxidants

## Abstract

The formation of advanced glycation end-products (AGE) in tissues is a physiological process; however, excessive production and storage are pathological and lead to inflammation. A sedentary lifestyle, hypercaloric and high-fructose diet and increased intake of processed food elements contribute to excessive production of compounds, which are created in the non-enzymatic multi-stage glycation process. The AGE’s sources can be endogenous and exogenous, mainly due to processing food at high temperatures and low moisture, including grilling, roasting, and frying. Accumulation of AGE increases oxidative stress and initiates various disorders, leading to the progression of atherosclerosis, cardiovascular disease, diabetes and their complications. Inborn defensive mechanisms, recovery systems, and exogenous antioxidants (including polyphenols) protect from excessive AGE accumulation. Additionally, numerous products have anti-glycation properties, occurring mainly in fruits, vegetables, herbs, and spices. It confirms the role of diet in the prevention of civilization diseases.

## 1. Introduction

Glycation is a complex process involving numerous steps, resulting in the formation of advanced glycation end-products (AGE) [1]. AGE may be divided into endogenous and exogenous [2]. Endogenous includes premelanoidins and melanoidins, whereas exogenous AGE mainly occur in food and cigarette smoke [3]. The formation of AGE in the body is a physiological process; nevertheless, excessive production and accumulation lead to inflammation [4]. Furthermore, increased glucose availability among patients who suffer from diabetes enhances AGE production, causing chronic complications [5]. The amount of AGE is also associated with the intake of food, particularly heat-treated products, such as roasted and grilled, which leads to the formation of new AGE [1]. Interestingly, the assessment of AGE in the skin may be a helpful diagnostic biomarker of cardiovascular disease [6], and researchers also investigate the receptor of AGE—RAGE, receptor for advanced glycation end products—due to the difficulties in achieving reproducibility of the results of serum AGE [7].

In homeostasis, RAGE expression is insignificant in neurons, smooth muscle cells, monocytes and fibroblasts [8]. In contrast, its expression is higher during cell activation (during stress or inflammation) and linked to increased RAGE ligand [9]. Scavenger receptors located on monocytes/macrophages remove AGE by endocytosis [10]. The scavenger receptors of AGE class A (SR-A) may bind ligands with negative surface charge (e.g., acetylated and oxidized LDL) [11]. One of the scavenger receptors of AGE is CD36, which catches ligands of fatty acids, collagen and oxidized LDL (low-density lipoproteins). Furthermore, the number of receptors is higher in blood vessels affected by the atherosclerotic process, causing the transformation of macrophages into foam cells, which enhance the atherosclerosis process [12]. Galectin 3, lectin and AGE receptors may affect splicing mRNA [13], whereas lectin gene removal causes increased accumulation of AGE in kidneys, accelerating the destruction of the renal glomeruli [14]. Additionally, AGE affects the body’s defense function through binding free proteins, such as lysozyme and lactoferrin [15,16]. Melting of ligand and RAGE leads to activation of kinase MAP, which takes part in various signaling pathways and increases the formation of reactive oxygen species due to oxidase NADPH activation [17]. Among patients with diabetes, AGE-RAGE complexes cause a higher production of VCAM-1 (vascular cell adhesion molecule 1), ICAM-1 (intercellular cell adhesion molecule-1), pro-inflammatory interleukins and growth factors [18]. The effect of protein glycation end-product accumulation on cellular mechanisms is detailed in Figure 1. 

Blocking the interaction between ligand and RAGE might reduce AGE’s toxic effect, affecting the effects of the therapy [8]. It is vital to notice that glycation including glucose is a lengthy process, although fructose, glucose-6 phosphate and intracellular glucose may create AGE relatively quickly [19]. Glycation is a non-enzymatic process; thus, its speed is expressed by the serum level of glucose (fructose, galactose) [20,21].

The association between the way and process of collecting AGE are not well known. The role of AGE in the pathomechanism of various diseases, such as cardiovascular diseases [22,23], diabetes and complication of them [6,24,25], dyslipidemia [26,27], obesity [28,29] have been confirmed in many clinical trials. However, an association between diet and level of AGE in the body has been found, indicating culinary techniques which increase the amount of them in food and present products’ groups with higher quantity [5,30,31].

## 2. AGE Accumulation and Metabolic Diseases

Atherosclerosis is a slowly progressing disease, and its symptoms depend on the localization of the atherosclerotic plaque. Clinical symptoms of atherosclerotic plaque manifest in older age, although plaque formation may commence in younger age [32]. The accumulation of AGE is observed in the atherosclerotic plaque of the aorta, causing loss of elasticity of the vessel, particularly among patients with diabetes, hypertension and kidney diseases [33]. Additionally, the AGE level affects the stage of coronary artery disease [34]. However, there is a number of mechanisms linking glycation and atherosclerosis [35]. AGE may bound with specific receptors, such as the most common, RAGE, which carries membrane signals, along with the galectin 3 receptor, scavenger receptors, etc. [31]. Melding AGE and RAGE causes oxidase NADPH activation, leading to oxidative stress in various cells [3], which subsequently results in a cascade of inflammatory processes in the arterial walls due to the active activity of macrophages and thrombocytes, causing thrombosis [22,23]. Inflammatory processes are inextricably linked with the activation of transcription factors, including NF-kB factor associated with forming reactive oxygen species [36]. In addition, it activates several pro-inflammatory genes: interleukin genes, interferon β and γ genes, monocyte proteins, stimulants: granulocytes (G-CSF), macrophages (M-CSF) and granulocytes-macrophages (GM-CSF). It is also responsible for activating the genes associated with growth, differentiation and adhesive molecules. The accumulation of AGE compounds is accompanied by a decrease in nitric oxide (NO) availability, which has anti-proliferative and anti-inflammatory properties [37]. Nitric oxide is inactivated to peroxynitrite by the action of reactive oxygen species. The reactive forms of nitrogen formed in this way can convert LDL, the main contributors to atherosclerosis, into a form which easily migrates to the vessel wall: NO2-LDL [23]. AGE products also influence the activation of monocytes, leading to a significant expression of the CD36 receptor, which has a strong affinity for AGE, as a result of which lipoproteins are taken from the blood, foam cells are formed, and, consequently, the atherosclerotic disease progresses [38]. Glycation products activate reactive oxygen species in thrombocytes, hence increasing the release of prostanoids, inhibiting prostacyclin production, and enhancing the action of the plasminogen activator inhibitor PAI-1 [39]. It results in the aggregation of thrombocytes, inhibition of clot dissolution and stabilization of fibrin, which causes the formation of blood clots in places of damaged tissue. A large number of AGE products in the vessels is associated with the destruction of the extracellular matrix [23]. Nevertheless, the pathogenesis of atherosclerosis and other cardiovascular diseases may also be related to endothelial dysfunctions caused by circulating endothelial progenitor cells (EPC), which do not fulfil their physiological function and do not repair the damaged endothelium [40]. Glycation end products contribute to this pathology, as the integrin-binding sequence (RGD motif) in fibronectin is glycated, which greatly reduces EPC adherence and migration. In addition, RAGE may also trigger the programmed death of progenitor cells [41]. AGE affects the cells of osteoblasts’ outer membrane, which leads to calcification in glycated sites [42]. In fact, the formation of calcium deposits in the atherosclerotic plaque area occurs in the advanced stages of atherosclerosis and affects the occurrence of myocardial infarction [43,44]. The accumulation of AGE in the body affects the success rate of coronary revascularization in individuals with diabetes [45]. Additionally, increased serum AGE levels have also been associated with higher coronary heart disease mortality in persons with type 2 diabetes [46]. Elevated levels of glycation end products in the blood of patients with type 1 diabetes have been associated with cardiovascular events, which in some cases have been fatal, regardless of whether there were cardiovascular risk factors (age, BMI, smoking, arterial hypertension and hyperlipidemia) [22,45]. In addition, the increased concentration of AGE in the skin in persons with type 1 diabetes is associated with the thickening of the intima/media complex and the development of microvascular complications [47]. In individuals with diabetes, elevated AGE levels lead to the accelerated development and progression of heart failure both indirectly through vascular effects and significant direct effects on the myocardium via cross-linking of extracellular proteins, as well as directly through AGE receptors in the heart tissue [45]. In a study by Kilhovd et al., the mean concentration of AGE in the blood serum of patients with cardiovascular disease and diabetes was significantly higher than in the control group [48]. In fact, the presence of protein glycation end products in the body is strongly associated with the persistent state of hyperglycemia [49]. The destructive effect of hyperglycemia is based on the activation of several metabolic pathways. The relationship of vascular complications and the non-enzymatic glycation of proteins was described in the 1980s by Brownlee. According to the study, by means of combining with the receptor on the macrophage membrane, AGE leads to an increased synthesis and release of pro-inflammatory cytokines, interleukin-1 and cachectin, i.e., tumor necrosis factor α (TNF-α). It has been proven that TNF-α acts autocrine on the macrophage and thus increases the expression of AGE receptors [50] which, in turn, results in stronger binding and endocytosis of AGE. Thus, faster elimination of glycation-modified erythrocytes can occur. Furthermore, AGE are also capable of modifying the extracellular matrix [51]. Microangiopathies are associated with modifying plasma proteins, morphotic elements, tissue proteins, as well as vessel walls. The process of non-enzymatic glycation of matrix proteins often leads to an increased susceptibility to coronary artery disease in persons suffering from both type 1 and type 2 diabetes [52,53,54,55]. Collagen is an example of matrix protein, which remains in the body for a prolonged period of time, and due to collagen glycation, fibrils and other solid structures of the body become stiff. However, the most dangerous effect of collagen glycation hinders the formation of three-dimensional collagen networks, which results in stiffening of the blood vessel walls and impeding blood flow [56]. However, patients suffering from type 1 and type 2 diabetes present a different course of this disorder [57]. 

The formation of advanced glycation end-products is also observed in the crystallin (protein) of the eye lens [58], which leads to conformational changes due to the formation of cross-links and causes the development of “macromolecular aggregates of crystallin”, absorbing an increased amount of light. The consequence of this process is cataract, i.e., a metabolic disorder of the lens, leading to transparency loss, which frequently occurs in individuals with diabetes [59]. Changes in the endothelium are also associated with both autonomic and peripheral neuropathy, which may lead to the development of diabetic foot syndrome [60,61,62]. The study by Vouillarmet et al. showed that the autofluorescence of the skin SAF (AGE concentration in the skin) correlated directly with the occurrence of the diabetic foot, and a correlation was observed between SAF levels and healing time in patients with diabetic foot syndrome [63]. Additionally, an increased glycation of platelet proteins and cell membrane dysfunction, increasing blood clotting, has also been observed in persons with diabetes [64]. Nevertheless, determination of the glycation of antithrombin III helps establish the degree of impairment of this protein’s function [65]. Glycation end products can be treated as a biomarker of diabetes, although the assessment of AGE concentrations by enzymatic methods is quite difficult. This is due to the high instability of AGE and the fact that their blunting is affected by various factors. It is possible to obtain more reliable results by measuring the receptors for AGE which circulate in the blood serum. Non-invasive methods for assessing the accumulation of advanced glycation products in the skin (SAF) are an alternative method and show a strong correlation with complications of diabetes [66]. Higher AGE levels have also been demonstrated in patients with type 1 diabetes, as well as concomitant factors in overweight or obesity [29]. The reduction in AGE level can be achieved by glycemic control and supplementing the diet with α-lipoic acid, vitamin A, as well as L-carnitine, with exercise also playing a significant role in the process [67,68].

## 3. Defense Mechanisms against Glycation

Defense mechanisms involved in the maintenance of AGE homeostasis include inborn defense system, enzymatic degradation, renal clearance, and cells degradation by receptors. Furthermore, skin pigmentation, metal redox chelation and structural conformation of enzymes, protecting reactive spaces also constitute protective factors, and the gut microbiome may be an additional element in the metabolism of exogenous AGE [69,70]. Enzymes responsible for the deglycation of protein include fructosamine 3-kinase, fructosamine oxidase, 2-oxaldehyde reductase and carbonyl reductase. In fact, they participate in the first or second step of the Maillard reaction, as well as remove dicarbonyl compounds. Enzymes degrade AGE systems of glyoxalases I and II, aldo-ketoreductases and aldehyde dehydrogenases, whereas glyoxalase I catalyzes the metabolism of dicarbonyl compounds and prevent binding of them with proteins, protecting them from AGE formation [71]. 

However, this mechanism is insufficient in patients suffering from diabetes [72]. Kidneys constitute a biological defense from AGE and the organs from which they are removed [73]. It is vital to bear in mind that the concentration of AGE in serum and tissues correlates positively with the stage of diabetic kidney disease [74]. Reducing AGE in food may significantly improve metabolic insulin response and decrease the risk of civilization diseases [75]. Fructosamine-3-phosphokinase catalyzes adducts with fructosamine, and together with lysine, they release one molecule of highly reactive 3-deoxyglucosone, which must be removed as soon as possible. Otherwise, it may react with residues of lysine, arginine and tryptophan, resulting in a re-glycation process [3]. Figure 2 presents the summary of AGE sources and their impact on the health status [5,75,76].

## 4. Nutrition and the Glycation Process

### 4.1. AGE Absorption through the Gastrointestinal Tract

After oral ingestion, only 10–30% of AGEs is absorbed [77], which can exist as single amino acids or proteins, or can be combined with other large as well as small proteins [78]. They can be absorbed by diffusion, e.g., Free CML can be absorbed by simple diffusion or be absorbed by the intestinal epithelium using peptide transporter 1 (PEPT1) [79,80]. Studies show that the absorption of some AGE is greater in the free form than in the protein-bound form [81] and their excretion in the feces is greater when bound to proteins [82] Forester’s study evaluated the effect of nutrition on the urinary excretion of Maillard reaction products, such as fructoselysin, pyralin and pentosidine. As a result of a diet free of the end products of protein glycation, the urinary excretion of free pyralin and fructoselysin, which was calculated by furosine analysis, was reduced by about 90%. In contrast, the excretion of pentosidine was reduced by about 40%. The excreted pentosidine was 60% from the free form from the coffee consumed, and 2% from the peptide-bound amino acid consumed from the roasted product. The authors of the study concluded that the bioavailability of free pentosidine was better compared to the protein-bound form [81]. Conversely, the Roncero-Ramos study investigated the intake and excretion of carboxymethyl-lysine (CML) after feeding rats using products rich in advanced glycation end products (AGEs). It was shown that fecal excretion of CML was related to the concentration of CML in food products. Moreover, fecal excretion of CML was the main route of CML excretion [82]. In addition, the relationship was found between the amount of free CML in plasma and urine, and the amount of CML ingested in the diet. In fact, some authors suggest that AGEs bound to plasma proteins are formed by endogenous processes [83].

However, certain AGEs are not absorbed in the gastrointestinal tract; thus, the unabsorbed AGEs pass into the colon where they can be metabolized by the microbiome. AGE absorption through the gastrointestinal tract is presented in Figure 3.

### 4.2. Products Containing Increased Amounts of Age

Advanced protein glycation products in food are formed in the course of the Maillard reaction, as well as by means of interactions between oxidized lipids and proteins, as a result of which carboxymethylysine can be formed [84]. Knowing the factors which determine the amount of AGE in food, patients do not have to immediately quit or limit their favorite foods, and they can focus on changing the culinary techniques. It is the method of cooking, not the actual composition of the food, that determines the final content of the AGE. For instance, stewing or steaming meat generates much smaller amounts of AGE as opposed to baking or frying. Table 1 presents AGE contents in the selected food products [80].

Studies introducing a low-AGE diet indicated that healthy individuals, both young and older, with diabetes but without CKD (Chronic Kidney Disease), as well as patients with CKD but without diabetes, responded to this intervention with a significant decrease in the circulating AGE [85,86]. In the study Semba et al., the authors also noted that a decrease in serum AGE concentrations was accompanied by a simultaneous reduction in inflammatory markers, oxidative stress and endothelial dysfunction. Moreover, in patients with insulin resistance, this diet improved insulin sensitivity according to the HOMA-IR (*Homeostatic Model Assessment—Insulin* Resistance) index [87]. Nevertheless, food sources containing AGE may vary depending on the country. The largest database created in New York, indicates products with the highest content of carboxymethylelysine measured by the ELISA method. It was found that the average consumption of AGE in the diet of healthy persons from the New York area is about 14,700 +/− 680 kilo-unit AGE per day. Therefore, it is possible to estimate whether the consumption of AGE in a given patient’s food is greater or less than 15,000 kilo-units [31]. In addition, many studies have shown that exogenous AGE can be found in a wide range of products, which include the aforementioned biscuits, cookies, but also bread, peanut butter, or processed meat [84]. The high content of AGE is also present in cereals, ice cream and soft drinks with corn syrup, which contains large amounts of fructose [88,89].

### 4.3. Urbanization and AGE Accumulation

Over the years, the demand for foods which can be prepared in a quick and convenient way has increased. The modern lifestyle has led to a significant development in the ready-made and highly processed food industry, which is characterized by an excessive content of exogenous AGE [5]. In contrast, home-prepared meals contain less AGE [90]. In the majority of cases, AGE derived from processed foods are produced during dry-heat technology [70], which in turn accounts for the high level of AGE in ready-made cookies, biscuits or chips. Furthermore, such products often contain more than one ingredient which promotes the production of AGE, i.e., cheese, nuts, saturated and/or trans fats [5]. Nevertheless, more and more children and infants consume these products, which is a cause of concern, since the likelihood of developing diabetes, kidney failure, heart disease and dementia increases due to the excessive load of exogenous AGE [91,92,93,94]. During the study on an animal model, Cai et al. have shown that age-related dementia may be related to AGE in foods, in particular to methylglyoxal [95]. Glyoxal, methylglyoxal and 3-deoxyglucosone are classified as by-products formed during baking, which are also found in carbonated drinks containing corn syrup, characterized by a high content of fructose, as well as in some fermented products, e.g., in beer, or wine [96,97]. The highest levels of exogenous AGE in foods of animal origin are observed in beef and ripened cheeses, e.g., in parmesan. This group of products is followed by poultry, pork, fish and eggs [98]. High-fat spread products, such as butter, margarine, cream cheese, and mayonnaise, also contain large amounts of AGE, due to the heat extraction and purification processes involved in their production [99]. Through high-temperature food processing techniques, flavoring and coloring substances are created, found in such foods as roasted coffee, cocoa, cereals, during baking or grilling. These substances significantly affect the perceived taste and smell of dishes subjected to the aforementioned heating processes [100]. As has already been mentioned, the basic mechanism for the formation of these substances is the Maillard reaction. The rate of this reaction is accelerated by an increase in temperature, which is also affected by the pH value [101]. Therefore, pickling will have an AGE-lowering effect due to the reduction in the local pH of a given food (for instance with the use of lemon juice or vinegar). In turn, a high pH value (maximum pH of 10) intensifies the formation of AGE, due to the fact that amino groups of proteins are in alkaline form in this state, and sugars in a reducing or open-chain form, which increases their reactivity [98]. In the study, Scheijen et al. it was shown that the highest content of carboxymethylelysine (CML) (one of the representatives of the final products of protein glycation) had peanut butter, chocolate sprinkles and black pudding. In addition, some of the higher AGE values were observed in canned meat, cereal preparations processed at high temperatures, butter and roasted coffee [30]. It is worth noting that the lowest values of exogenous AGE are found in vegetables and fruits, and low concentrations of CML were also detected in dairy products. High-fat products such as oils, olive oils are also characterized by a small amount of AGE [98]. Cooking, steaming and stewing without frying reduces the formation of the final products of protein glycation in food products, as opposed to when subjected to high temperatures [30]. However, in order to be able to estimate the overall content of glycation end products in the diet, a larger number of individual AGE should be involved in the studies using instrumental analyses; thus, it is necessary to conduct further research [102].

## 5. Limiting the Amount of AGE in the Body—Dietary Management

Many studies are conducted to detect substances which would block the formation and accumulation of AGE products in the tissues of the body, or affect the already formed products of advanced glycation [103]. In addition to the diet-related modifications, the formation of protein glycation end products is also affected by proper treatment of metabolic diseases such as diabetes, hypertension, obesity, dyslipidemia and gout. At the moment, experts are considering three therapeutic aspects [104,105]. The first concerns the use of oligo- or polyamine compounds, which have the ability to compete with amino groups of proteins for the attachment of reactive carbonyl groups. Another issue is the use of substances which would destroy protein cross-linking products, the so-called AGE-breakers. Furthermore, the third therapeutic aspect refers to supplementing the diet with compounds which reduce oxidative and carbonyl stress, i.e., free radical scavengers. The first group of medications includes hydrazine derivatives, i.e., aminoguanidine, which have not passed clinical trials successfully, although research on its derivatives is currently being conducted. However, the side effects, including the generation of autoantibodies, flu-like symptoms and anemia, have prevented the widespread use of aminoguanidine [106,107].

It can bind competitively to carbonyl groups of reducing sugars and thus protect “lysine and arginine residues of body proteins from their modification. Moreover, drugs decreasing blood glucose levels, such as metformin, have a similar effect [104,105,108]. The effectiveness of inhibitors of AGE formation, such as Pyridoxamine (PM), has been confirmed in clinical trials in patients with diabetes [109].

In recent years, research has been focused on substances with antioxidant properties, which are common in the plant world, e.g., in the form of polyphenols. More and more studies indicate a link between plant extracts rich in polyphenols and anti-glycation properties [110]. Polyphenols perform a number of functions in plants, e.g., antioxidative, dyeing, antifungal and anti-insecticides, anti-radiation [111], and they have been classified according to the structure of the carbon skeleton [112]. They have various beneficial effects on the body, e.g., antioxidant, cardioprotective, diuretic, antitumor, and anti-inflammatory [113]. The cardioprotective effect of polyphenols may be due to the so-called French paradox, according to which in France there is a much lower incidence of cardiovascular events, compared to the population of other Western European countries, despite the consumption of a large amount of animal fats and a high rate of cigarette smoking [114,115]. This interesting phenomenon is explained by the regular consumption of products rich in various polyphenols, including resveratrol. The phenomenon of the French paradox demonstrates that a diet based on a large amount of plant products rich in antioxidant compounds may constitute an effective way to prevent civilization diseases, including complications associated with diabetes [116]. Interestingly, researchers from Madras studied how green tea extract affected the formation of the final products of protein glycation in collagen taken from diabetic rats. The results showed that in the study group, blood glucose concentrations almost doubled, compared with the control group (rats with diabetes, not fed with green tea extract). The anti-glycation properties of green tea may be due to the presence of polyphenols (e.g., caffeine, epigallocatechin, epigallocatechin gallate and epicatechins) [117,118]. Another study demonstrated the anti-glycation properties of guava extract. According to the in vitro studies, this extract reduced the content of α-dicarbonyls by more than 95%, as well as the presence of AGE by approximately 70% (determined by the fluorescent method). Polyphenolic compounds found in guava extract include, e.g., gallic, ferulic, caffeic and chlorogenic acids, and also comprise flavonoids, such as rutin, quercetin and kaempferol [119]. In the study Noowaboot et al., anti-glycation properties of white mulberry extract have been demonstrated. A significant reduction in blood glucose levels was also observed in diabetic rats in which this extract was administered. Glycated hemoglobin levels in rats in the study group were also decreased (by approximately 7%) compared to rats which did not receive the extract [120]. The same authors also showed that white mulberry extract inhibits the processes of glycation and the formation of AGE. The polyphenols present in the fruits and leaves of white mulberry include such substances as 3-(6-malonylglycoside) quercetin, 3-(6-malonylglycoside) kaempferol, rutin, tannin, coumarin, astragaline, isoquercitrin, phenolic acids and terpenes [120]. Additionally, lemon balm contains rosemary acid, which inhibits the formation of AGE compounds. In vitro studies demonstrated that lemon balm extract significantly reduced the intensity of protein fluorescence in a solution of bovine serum albumin (BSA) and glucose [121]. In another study, the extract was shown to have a stimulating effect on glucose uptake in adipocytes, which also promotes the treatment of metabolic disorders associated with AGE concentration in the body [122].

Yerba mate is becoming increasingly appreciated in various countries due to its numerous beneficial properties. It is said that it may contain more polyphenols than red wine or green tea. According to the research, it has antioxidant, anti-inflammatory and anti-glycation properties, and it contains vitamins A, B, C and E, tannins, chlorogenic acid, caffeic acid, oleic acid and catechins. In vitro studies showed a significant decrease in AGE compounds in BSA solutions, which were incubated with methylglyoxal and yerba mate extract [123,124]. Anti-glycation properties have also been found in mountain ash, thyme, black currant, apples and many other vegetables and fruits [125,126]. In fact, more than eight thousand polyphenolic compounds have been described up to date, with a daily recommended consumption of about 0.5–1 g. It is worth emphasizing the important role played by a diet rich in vegetables and fruits, which constitute a source of numerous polyphenols, in the context of treatment and prevention of complications as a result of persistent hyperglycemia [103]. Interestingly, both in vivo and in vitro studies have also revealed that curcumin can inhibit the formation of AGE. It can be considered as an inhibitor of the glycation process due to the antioxidant properties it exhibits. A diet with a curcumin content of 0.5% results in a decrease in blood and urine glucose levels, to a degree similar to aminoguanidine, which is a standard glycation inhibitor. Moreover, it was observed that the administration of curcumin to the diabetically induced rats resulted in a significant reduction in blood glucose, glycated hemoglobin and reduced insulin resistance. El-Moselhy et al. demonstrated that administration of curcumin at a dose of 80 mg/kg body weight decreased TNF-α levels, which counteracted hyperglycemia and reduced insulin resistance in rats on a high-fat diet [127]. In fact, the addition of 0.1% curcumin to the diet rich content of AGE compounds (study on mice) led to the normalization of the concentration of pro-inflammatory and anti-inflammatory cytokines, chemokines, carboxymethylelysine, C-reactive protein and HbA1c. Furthermore, curcumin also reduced blood glucose levels, lipid peroxidation and increased activity of antioxidant enzymes and increased the diameter of pancreatic islets in mice with induced diabetes. Curcumin also affected the regeneration of pancreatic islets and improved the functioning of β cells in vitro. Notably, it suppressed inflammatory processes in the body and prevented apoptosis of secretory cells located on pancreatic islets. In the study of Gutierres et al., researchers have proven that the administration of yoghurt with curcumin normalized blood glucose levels, the activity of aspartate and alanine transaminases, and also reduced proteinuria in animals with induced diabetes [128]. In addition, a Mediterranean diet enriched with curcumin for a period of 3 months significantly reduced the amount of AGE and RAGE in the blood serum of subjects who exercised intensively [129]. It is worth bearing in mind that curcumin can also prevent diabetic complications and studies indicate that when administered intragastrically at a dose of 150 mg/kg body weight/day, it prevented inflammation and fibrosis of the heart muscle. It also inhibits apoptosis of cardiomyocytes, renal cells and mesangial tubules. Curcumin-polyphenols has a neuroprotective effect and prevents the hyperglycemia-associated changes in the cerebellum. It has a protective effect on the retina of the eye, as it prevents its thinning and prevents disorders related to the functioning of photoreceptors. Diabetic complications are often accompanied by non-alcoholic steatohepatitis, which can result in fibrosis. In the study by Tang and Chen curcumin has been shown to have an inhibitory effect on the activation of stellate cells in the liver, as a result of glycation products found there, thus preventing fibrosis [130]. The numerous health-promoting properties of curcumin indicate its wide use as a therapeutic agent in the course of diabetes, hyperglycemia, insulin resistance, in the regeneration of islet cells, and as a source of antioxidant and anti-inflammatory properties. Numerous studies have confirmed the effectiveness of curcumin use in the daily [131]. In the study Rodríguez J.M. et al. the authors observed that calorie restriction and the Mediterranean dishes (based on vegetables, legumes, fish, whole grains, olive oil and fermented milk products, excluding red wine), led to a number of positive changes in women with obesity before menopause. The applied interventions positively affected the adipose tissue, the level of serum CML and led to a decrease in lipoproteins. Furthermore, the previous studies demonstrated that products such as breakfast cereals, ready-made sweets and salty snacks, powdered milk, processed cheeses contain significant amounts of exogenous AGE, and the age of their consumers is continuously decreasing. The researchers also measured the level of CML and AGE in the most consumed foods and proved that the study population was exposed to an increased intake of gliotoxins in the diet. Therefore, even though the Mediterranean diet is considered a low-AGE diet, it is not always the rule. When summarizing this study, the researchers concluded that lowering serum CML was achieved by an effectively reduced calorie intake, while favoring Mediterranean cuisine products. They did not indicate that lowering the consumption of foods containing elevated amounts of AGE could significantly affect the study results [132].

## 6. Strategies to Reduce the Impact of AGEs on the Body

In addition to the exogenous food sources of AGEs, endogenous formation of AGEs also plays a considerable role The main strategies targeted to reduce the accumulation of protein glycation end products in the body include the abovementioned lifestyle interventions, glycemic control especially in diabetic patients. Natural defense mechanism and pharmacological strategies and AGEs inhibitors are also involved in this process (Figure 4).

However, regardless of the formation route, AGEs constitute a major pathogenic factor in chronic vascular complications of diabetes and exacerbate dementia in the elderly. Cellular dysfunction may be related to their action on the RAGE receptor or the extra-receptor pathway. The main cause of AGE formation and accumulation is carbonyl stress, associated with the excessive production of reactive dicarbonyls or their reduced utilization by the glyoxalase system or endogenous scavengers [133]. In contrast, the pathogenic role of AGEs in tissue damage in the inflammatory foci is played by the production of glycolaldehyde by myeloperoxidase from the activated macrophages and neutrophils [134]. The absolute reduction in AGE synthesis and delivery is essential to reduce the incidence of metabolic disorders. Studies have shown the effect of various drugs and substances, such as the PPARγ inhibitor rosiglitazone [135] or MAPK inhibitor on reducing tissue AGE accumulation. As a lipid lowering medications, Statins (Atorvastatin and simvastatin) also play an important role in process of inhibiting AGE formation due to their anti-oxidative properties [136]. Moreover, zinc which shows similar properties, has also been considered as a method of reducing the effects of AG-Es on cells; thus, zinc supplementation may inhibit AGE formation. Additionally, it also inhibits AGE-induced cell apoptosis and the formation of protein carbonyls, possibly through various signaling pathways [137]. In a study by Shah et al., the antiglycation effect of a rinacanthin-rich extract was investigated in comparison with the markers rinacanthin-C (RC), rinacanthin-D (RD) and rinacanthin-N (RN). An anti-glycation test indicated that it exhibits nearly equivalent glycation inhibitory activity as R, which can be used in the prevention of Age concentration in the body and even in the treatment of chronic diseases, such as diabetes and senile dementia. In another study, the extract was shown to have a stimulating effect on glucose uptake in adipocytes, hence facilitating the treatment of metabolic disorders associated with AGE concentration in the body [138].

Summing up, no clear treatment has been found to reduce AGEs in healthy individuals yet. Nevertheless, there are various compounds presenting potential AGE-reducing effects, and a healthy, unprocessed diet, rich in antioxidant substances also has a beneficial effect on reducing AGE accumulation in the tissues.

## 7. Conclusions

AGEs have a pro-inflammatory effect and promote oxidative stress, which negatively affects tissues in the human body. Numerous studies indicate that the end products of glycation play a vital role in the pathogenesis of various diseases, including cardiovascular diseases, diabetes and diabetes-associated complications, dyslipidemia, and also faster skin aging. However, the defense mechanisms protect the body from an excessive amount of harmful AGE and involve, e.g., innate antioxidant buffer systems, enzymatic systems including catalases, peroxidases and glutathione reductases. A large role in the anti-glycation effect is also attributed to polyphenolic compounds widely distributed in plant organisms. Defense mechanisms also include DNA repair mechanisms, the gut microbiome, skin pigmentation, etc., the common goal of which is to maintain redox balance. It is worth bearing in mind the heat treatment of prepared meals, since it is mostly accountable for the formation of harmful AGE. This issue is particularly significant in terms of the individuals with specific dietary needs, where the selection of appropriate foods and the preparation method are crucial. Therefore, nutritional education of patients is vital, since this knowledge constitutes an indispensable factor in the successful treatment.

## Figures and Tables

**Figure 1 nutrients-14-03982-f001:**
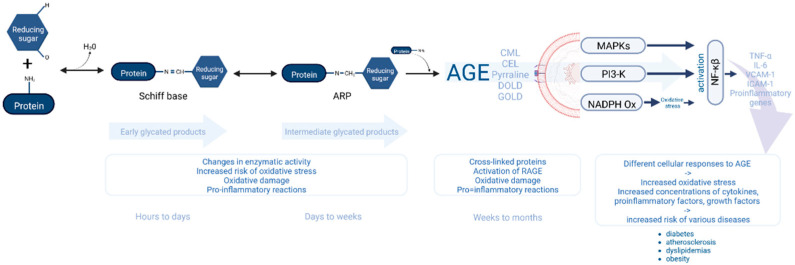
The mechanism involved in the relationship between the accumulation of protein glycation end products and cellular defense systems.

**Figure 2 nutrients-14-03982-f002:**
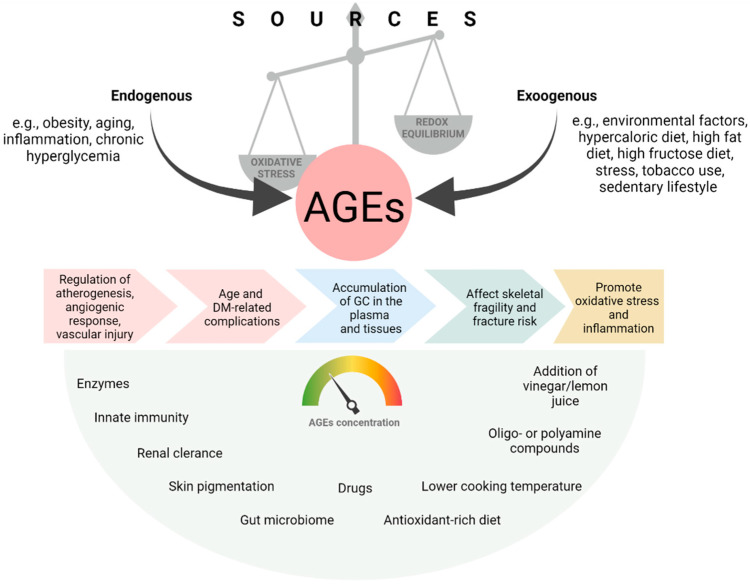
AGE sources, their impact on the human body and on lowering and defense factors. AGE—advanced glycation-end products; GC—glycation compounds.

**Figure 3 nutrients-14-03982-f003:**
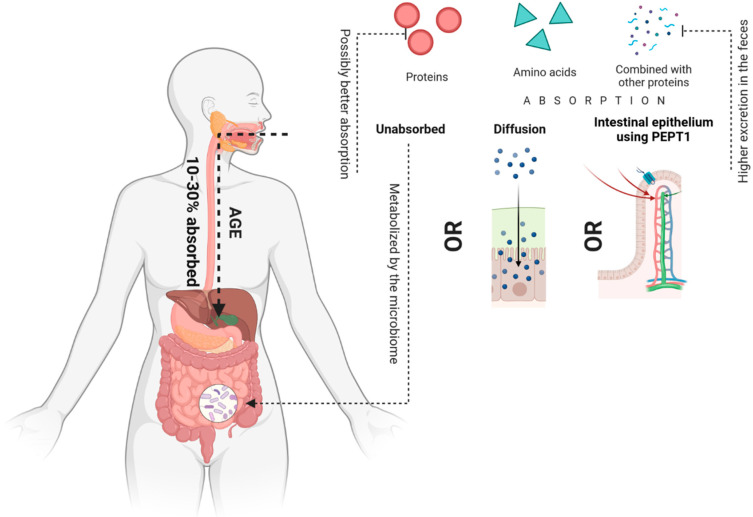
AGE absorption through the gastrointestinal tract.

**Figure 4 nutrients-14-03982-f004:**
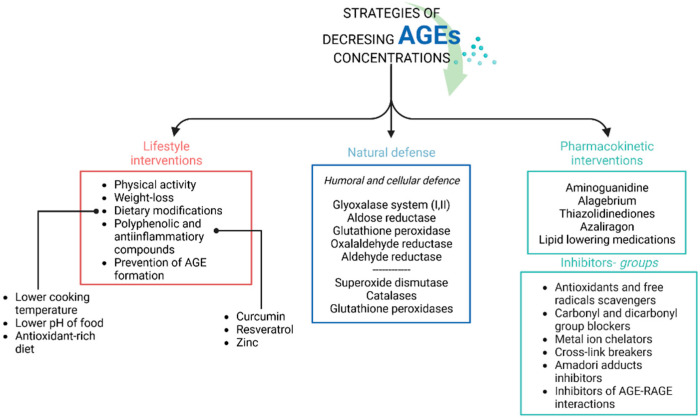
The main strategies aiming to reduce the accumulation of protein glycation end products in the body.

**Table 1 nutrients-14-03982-t001:** AGE contents in the selected food products [1,31,84,85].

High Level of AGE	Low Level of AGE
Foods rich in protein	Low-fat products
Foods rich in fat	High-carbohydrates products
Baked and grilled food	Raw products
Fraying products	Products cooked in low temperature
Animal products	

## Data Availability

Not applicable.

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
