# Peer review of "Accumulation of Advanced Glycation End-Products in the Body and Dietary Habits"

_nutrients, 2022, doi:10.3390/nu14193982_

Round 1

Reviewer 1 Report

In this paper, the authors summarized the accumulation of advanced glycation end-products in the body and the influence of individual nutritional habits. First, the authors introduced the two sources of glycation: endogenous and exogenous, leading to the relationship between glycation homeostasis and diet in vivo. Then, the authors presented the link between glycation levels and metabolic diseases, especially atherosclerosis. Next, the authors described the mechanisms of the body's defense against the glycation process, including innate defense systems, enzymatic cleavage, renal clearance, and receptor degradation of cells, among others. Finally, the authors highlighted the link between nutrition and the glycation process, including the amount of exogenous glycation introduced by food, the way in which the body absorbs glycation, and the way in which dietary management can limit the level of glycation in the body. Overall, the authors' review on nutrition and glycation is relatively comprehensive and can provide some degree of nutritional guidance to patients, which is perhaps more important in slowing down the progression of the disease.

However, I think the following issues need to be revised or further clarified.

(1)   Lines 209-211, Studies show that the absorption of some AGE is greater in the free form than in the protein-bound form and their excretion in the feces is greater when bound to proteins. What are some AGEs and can you write specific forms (CML? CEL? Or others)? What does absorption in free form mean? Is it the absorption of carbonyl small molecules and what is the relationship between absorption and glycation?

(2) The amino acid diagram on the upper right in Figure 2 showed, why was A B S O R P T I O N chosen? As far as I know, advanced glycation is mostly modified on R or K. Are there any other of these amino acids that have been reported as glycation modification sites?

(3) Line 291 states At the moment, experts are considering three therapeutic aspects. I think that it would be easier for the reader to understand the following three categories if a comprehensive diagram is provided.

(4) There seems to be a problem with the full name of BSA in line 338, beef serum albumin (BSA), please confirm if it should be bovine serum albumin.

Reviewer 2 Report

The authors have presented an update informtion regarding accumulation of advanced glycation end-products in the body and nutritional habits, however they need to imrprove the manuscript as follows:

1. The English language should be improved with a professional copyeditor.

2. Introduction and literature review must be improved by citing some relevant recent information such as:

https://www.ncbi.nlm.nih.gov/pmc/articles/PMC5701406/

https://www.ncbi.nlm.nih.gov/pmc/articles/PMC5822505/

3. Improve the qulaity of figures and add up some mechanistic figures

4. Add a section about recent strategies for controlling AGEs

Round 2

Reviewer 2 Report

Accepted in current form